# Postoperative Changes in Pulmonary Function after Valve Surgery: Oxygenation Index Early after Cardiopulmonary Is a Predictor of Postoperative Course

**DOI:** 10.3390/jcm10153262

**Published:** 2021-07-23

**Authors:** Tomohiro Murata, Motohiro Maeda, Ryosuke Amitani, Atsushi Hiromoto, Makoto Shirakawa, Masaru Kambe, Yuji Maruyama, Hajime Imura

**Affiliations:** 1Department of Cardiovascular Surgery, Nippon Medical School Musashikosugi Hospital, Kawasaki 211-8533, Japan; t-murata@nms.ac.jp (T.M.); m-maeda@nms.ac.jp (M.M.); rosuke620@nms.ac.jp (R.A.); atsu.hiromoto@gmail.com (A.H.); s5051@nms.ac.jp (M.S.); maruyamayuji@nms.ac.jp (Y.M.); 2Tokyo Heart Center Osaki Hospital, 5-4-12 Kitashinagawa, Shinagawa-ku, Tokyo 141-0001, Japan; sean.valentine.0214@gmail.com

**Keywords:** cardiopulmonary bypass, lung injury, valve surgery, prolonged ventilation, early extubation

## Abstract

Objective: To determine pulmonary functional changes that predict early clinical outcomes in valve surgery requiring long cardiopulmonary bypass (CPB). Methods: This retrospective study included 225 consecutive non-emergency valve surgeries with fast-track cardiac anesthesia between January 2014 and March 2020. Blood gas analyses before and 0, 2, 4, 8, and 14 h after CPB were investigated. Results: Median age and EuroSCORE II were 71.0 years (25–75 percentile: 59.5–77.0) and 2.46 (1.44–5.01). Patients underwent 96 aortic, 106 mitral, and 23 combined valve surgeries. The median CPB time was 151 min (122–193). PaO_2_/FiO_2_ and AaDO_2_/PaO_2_ significantly deteriorated two hours, but not immediately, after CPB (both *p* < 0.0001). Decreased PaO_2_/FiO_2_ and AaDO_2_/PaO_2_ were correlated with ventilation time (r^2^ = 0.318 and 0.435) and intensive care unit (ICU) (r^2^ = 0.172 and 0.267) and hospital stays (r^2^ = 0.164 and 0.209). Early and delayed extubations (<6 and >24 h) were predicted by PaO_2_/FiO_2_ (377.2 and 213.1) and AaDO_2_/PaO_2_ (0.683 and 1.680), measured two hours after CPB with acceptable sensitivity and specificity (0.700–0.911 and 0.677–0.859). Conclusions: PaO_2_/FiO_2_ and AaDO_2_/PaO_2_ two hours after CPB were correlated with ventilation time and lengths of ICU and hospital stays. These parameters suitably predicted early and delayed extubations.

## 1. Introduction

Impaired pulmonary gas exchange is a common complication after cardiopulmonary bypass (CPB) [1,2]. The impaired gas exchange lasts for several days after surgery with only modest improvements [3], and patients are weaned from mechanical ventilation under such conditions. On the other hand, fast-track anesthesia is now widely used and the benefits of early extubation are well-recognized in cardiac surgery [4,5]. In current practice, early extubation is attempted even for patients undergoing complex procedures. In such circumstances, assessments of post-CPB pulmonary function are essential for prediction of the postoperative course.

PaO_2_/FiO_2_ (PFR) is the ratio of arterial oxygen tension (PaO_2_) to the fraction of inspired oxygen (FiO_2_). PFR is frequently used as an indicator of pulmonary function; post-CPB lung injury (PCLI) is usually defined as a PFR of less than 300 [6,7]. Patients with PCLI have unfavorable postoperative outcomes in cardiac surgery [2,8]. Moreover, a recent study showed that a prolonged duration of mechanical ventilation (DMV) decreases mid-term survival rates in coronary artery bypass grafting (CABG) [9]. Thus, PCLI may be a key target for improving clinical outcomes in cardiac surgery. However, few studies have investigated how pulmonary dysfunction after CPB influences the postoperative course.

In general, complex valve surgery requires longer CPB than CABG, and may be associated with prolonged DMV [10,11]. We recently demonstrated that the early extubation rate was lower after valve surgery (approximately 44%) compared with the early extubation rate after relatively simple valve surgery and CABG [12]. CPB times for our valve surgeries were longer than the CPB times in previous studies [13]; therefore, pulmonary dysfunction after CPB might contribute to prolonged DMV. In fact, CPB time was an independent predictor of prolonged DMV in several previous studies [10,11].

During the last decade, we adopted fast-track cardiac anesthesia aimed at early extubation, even for patients undergoing complex valve surgery requiring long CPB times. In the present study, we hypothesized that PCLI significantly influences the postoperative course after current valve surgery and pulmonary function within a few hours after CPB predicts postoperative outcomes. This study provides new insights regarding post-CPB pulmonary function to improve perioperative management in cardiac surgery.

## 2. Patients and Methods

### 2.1. Study Approval and Patients

This retrospective study was approved by the institutional ethics committee and a waiver of consent was obtained. This study included 225 consecutive patients who underwent non-emergency aortic and/or mitral valve surgeries in our hospital between January 2014 and March 2020. All surgeries began in the morning. Fast-track anesthesia aimed at early extubation was employed by a single team in the perioperative period.

### 2.2. Anesthesia and Surgical Technique

Our standard anesthesia included midazolam (50–100 μg/kg), fentanyl (2–4 μg/kg), and rocuronium bromide (0.6 mg/kg) for induction of anesthesia. Continuous venous infusion of remifentanyl (0.2 μg/kg/minute), propofol (1–3 μg/mL of target-controlled infusion), and rocuronium bromide (7 μg/kg/minute) was used for the maintenance of anesthesia during surgery. Fentanyl (total of 10–20 mg) was added before sternotomy and closure of the chest.

All surgeries were performed through a median sternotomy and CPB was established with the ascending aorta and bicaval cannulations. Our standard protocol for CPB was as follows: (1) flow of 2.2–2.8 L/minute/body m^2^ and mean blood pressure of 50–70 mmHg; (2) rectal temperature of 36.0–37.0 °C; (3) urine flow of 1.5–2 mL/kg/h; (4) partial pressure of oxygen (PaO_2_) of 200–300 mmHg, carbon dioxide (PaCO_2_) of 35–45 mmHg, and pH of 7.35–7.45 under α-stat management; and (5) use of methylprednisolone, 30 mg/kg. Cardioplegic arrest was attained in all patients using antegrade and retrograde blood cardioplegia initially and every 30 min. Warm blood cardioplegia was used just before unclamping the aorta.

### 2.3. Ventilator Setting, Blood Gas Analyses, and Extubation Criteria

Mechanical ventilation was initiated immediately after the induction of anesthesia. The volume-control mode with the following settings was used: tidal volume, 7–10 mL/kg; respiration rate, 8–12/min; and FiO_2_, 0.4–0.6. Ventilator settings were changed to maintain PaO_2_ at ≥100 mmHg, partial CO_2_ pressure (PaCO_2_) at 35–45 mmHg, and pH at 7.35–7.45. The ventilator was stopped during CPB and restarted on weaning from CPB after full re-expansion of the lungs by several manual hyperinflations (peak pressure at 20–30 mmHg) and thorough intratracheal suctions. After CPB, the tidal volume and respiration rate were the same as described above until the first blood gas analysis (BGA) after CPB was available. After arrival in the intensive care unit (ICU), the ventilator was initiated with the same settings as in the surgical theater, and the patient was managed by the ICU doctors. BGA was performed at least 1–2 times an hour until extubation or within the first twelve hours. Patients were extubated when they met the following extubation criteria: (1) hemodynamic stability with reasonable urination and warm peripheral extremities; (2) PaO_2_ ≥ 75 mmHg and PaCO_2_ ≤ 50 mmHg with adequate spontaneous respiration under FiO_2_ ≤ 0.4 and end of expiratory pressure ≤ 5 cm H_2_O; (3) awake and able to respond to commands without new neurological symptoms; (4) no active bleeding with a reasonable change in hemoglobin and no requirement for volume replacement; and (5) no reasonable fear of re-intubation.

### 2.4. Data Collection and Statistical Analysis

All data were collected from medical records and deposited in the National Database system. BGA data before surgery and 0, 2, 4, 8, and 14 h after cessation of CPB were recorded. BGA was measured within 20 min of each time point.

Pulmonary function parameters were calculated as follows: PFR = PaO_2_/FiO_2_, alveolar–arterial oxygen gradient (AaDO_2_) = (760 − 47) × FiO_2_ − PaCO_2_/0.8 − PaO_2_, and respiratory index (RI) = AaDO_2_/PaO_2_ [14]. PCLI was defined as PFR < 300 measured 2 h after CPB cessation [2,6,7]. Extubation within 6 h after surgery was defined as early extubation, whereas DMV > 24 h was considered as prolonged.

Postoperative adverse events during the hospital stay included any death, heart failure requiring intravenous medical therapy (e.g., inotropes, carperitide, furosemide) beyond the fourth postoperative day, myocardial infarction, symptomatic stroke, acute renal failure requiring hemofiltration, major bleeding necessitating transfusion of red blood cell > 4 units and frozen fraction plasma > 4 units, sepsis, or major infection [4].

Differences between groups were evaluated using the Mann–Whitney U test for continuous variables and Fisher’s exact test for binary categorical variables. Univariate and multivariate logistic regression analyses were performed to identify predictors of prolonged DMV. The Hosmer–Lemeshow test was performed to evaluate the validity of the multivariate logistic regression model. The Pearson correlation coefficient was calculated to assess the relationship between pulmonary functions (PFR and RI) and postoperative course (DMV and length of ICU and hospital stay). Receiver operating characteristic (ROC) curves were used to evaluate the sensitivity and specificity for the effects of PFR and RI on DMV. All statistical analyses were performed using SPSS 25 J for Windows (IBM Corp., Armonk, NY, USA). A *p*-value < 0.05 was considered statistically significant for all analyses.

## 3. Results

### 3.1. Patient Characteristics and Perioperative Changes in Pulmonary Function

Baseline characteristics are shown in Table 1. PCLI was observed in 88 patients (39.1%). PCLI was common in males and patients with obesity, and chronic kidney disease. Patients with PCLI underwent more complicated and longer surgeries than patients without PCLI and had longer DMV and ICU and hospital stays (Table 2). During the observation period, no patient suffered from acute respiratory distress syndrome and no patients had steroid treatment.

Time-dependent changes in pulmonary functions are shown in Figure 1. PFR was significantly increased immediately after CPB in patients without PCLI, probably due to increased FiO_2_ (0.4; 0.4–0.6 vs. 1.0; 0.64–1.0, *p* < 0.0001, Appendix A). The increased PFR was not observed in patients with PCLI, although FiO_2_ was similarly elevated (0.4; 0.4–0.48 vs. 1.0; 0.6–1.0, *p* < 0.0001, Appendix A). Significant decreases in PFR were observed 2 h after CPB in patients with and without PCLI. PFR did not improve after 4 h. In contrast, RI consistently increased after CPB and patients with PCLI had consistently higher RI than patients without PCLI. PaCO_2_ levels in both groups were in a favorable range at each time point (Appendix A). Between 4 and 14 h after CPB, patients with PCLI were ventilated under higher FiO_2_ compared to patients without PCLI (Appendix A).

### 3.2. Univariate and Multivariate Analysis for Prolonged DMV

The univariate and multivariate analyses revealed that significant transfusion during surgery (*p* = 0.004, hazard ratio: 0.109, 95%CI: 0.024–0.449) and PCLI (*p* = 0.005, hazard ratio: 0.107, 95%CI: 0.023–0.506) were independent factors associated with prolonged DMV (Appendix A). The Hosmer–Lemeshow test confirmed the validity of this model (χ^2^ = 124.288 and *p* = 0.724).

### 3.3. Relationship between Post-CPB Pulmonary Function and Postoperative Course

As shown in Figure 2 and Figure 3, PFR and RI at two hours after CPB was correlated with postoperative DMV. PFR and RI also correlated with ICU and hospital stays. RI correlated better than PFR for each parameter (Table 3). Furthermore, the correlation coefficients for RI were higher when patients with adverse events were excluded (DMV: 0.517; ICU stay: 0.400; hospital stay: 0.220). As demonstrated in Table 3, 2 h after CPB, PFR and RI were correlated better with the postoperative course than at other time points.

### 3.4. ROC Analysis between Pulmonary Functions at Two Hours after CPB and DMV

PFR and RI 2 h after CPB were significant predictors of postoperative DMV (6, 12, 18, and 24 h) in the ROC analyses with an area under curve (AUC) between 0.772 and 0.853 (Figure 4 and Figure 5). The sensitivity and specificity were between 0.663 and 0.911 and 0.667 and 0.859, respectively. The predictive values of PFR and RI were almost equivalent (Figure 4 and Figure 5). PFR ≥ 354 (sensitivity: 0.80, specificity: 0.78) and RI ≤ 0.83 (sensitivity: 0.74, specificity: 0.82) predicted early extubation. PFR < 213 (sensitivity: 0.91, specificity: 0.67) and RI > 1.68 (sensitivity: 0.72, specificity: 0.86) also predicted prolonged DMV.

## 4. Discussion

The present study demonstrated that PCLI occurred frequently in current valve surgeries. Two hours after CPB, PFR and RI correlated well with postoperative parameters, such as DMV, ICU and hospital stays. RI correlated better than PFR in these analyses, but the predictive values (AUC) for DMV were similar. These correlations became stronger when patients with adverse events were excluded. Thus, this study shows, for the first time, that pulmonary function early after CPB is a useful predictor of postoperative course in current valve surgery.

Several previous studies have showed that CPB time is a significant predictor of DMV [10,11]; however, this was not confirmed in the present study. Our sub-analysis revealed that operation and CPB times were independent predictors of postoperative DMV (operation time: *p* = 0.007, CPB time: *p* = 0.032) only if PCLI was excluded from the analysis. Furthermore, CPB time was a strong predictor of PCLI (*p* = 0.022, hazard ratio 1.017, 95%CI: 1.002–1.031) in our sub-analysis. This is the first study to show that PCLI at two hours after CPB is a stronger predictor of postoperative course than CPB time.

PFR did not decrease immediately after CPB. PFR first decreased two hours after restarting mechanical ventilation and pulmonary perfusion. Although RI increased soon after CPB, major increases were observed in the next 2 h. Thus, impairment of pulmonary function peaked a few hours after CPB. This delay suggests that ischemia–reperfusion injury is a major cause of PCLI. Gasparovic et al. reported that pulmonary lactate release peaked 6 h after CPB and was correlated with AaDO_2_ [15]. Their study also demonstrated that pulmonary lactate release was intimately linked with the postoperative course. Another study revealed that PFR and RI were worse at 3–5 h after CPB than immediately after CPB; this changing pattern was similar even in patients with successful reductions in inflammatory lung injury by CPB [14]. These findings indicate that ischemia–reperfusion injury of the lungs is a major cause of post-CPB pulmonary dysfunction. This is in agreement with our previous study exploring ischemia–reperfusion injury in a pig model [16].

At 2 h after CPB, most patients were on a ventilator without spontaneous ventilation at similar FiO_2_ regardless of PCLI (Appendix A). We believe these relatively unified ventilation conditions helped reveal the correlation between PFR and RI 2 h after surgery. In fact, the ventilator conditions (FiO_2_ and spontaneous respiration rate) at 4 and 8 h were variable among patients with or without PCLI. On the other hand, pulmonary dysfunction did not occur immediately after CPB, and therefore, was not useful as a predictor. Weiss et al. reported that PFR 6 h after surgery was significantly correlated with postoperative DMV and length of hospital stay after CABG, whereas PFR at one and twelve hours was not correlated with postoperative DMV and length of hospital stay [17].

Patients with postoperative adverse events had significantly longer DMV (12.0; 6.0–17.0 vs. 26.0; 16.0–54.0, *p* < 0.0001). Interestingly, the postoperative adverse event rate was significantly higher in patients with PCLI than the adverse event rate in patients without PCLI. The increased adverse events may have contributed to the strong correlations of pulmonary function at 2 h with ICU and hospital stays. Several previous studies revealed that impaired gas exchange after CPB was linked to higher morbidities and mortality [2,8]. Thus, it is natural that PFR and RI are predictors of the postoperative course.

Patients with PCLI showed significant improvements in PFR and RI between two and four hours after CPB. No particular treatments for PCLI (such as steroids or hemofiltration) were used in this period; therefore, we believe that this improvement was mainly due to decreased atelectasis. Patients were already admitted into the ICU at this time, and patients with insufficient oxygenation underwent thorough physiotherapy. In previous studies, atelectasis was recognized as a major cause of pulmonary dysfunction in cardiac surgery [18,19]. Atelectasis was treated with simple physiotherapy [19] and the severity did not correlate with CPB time [20]. Even when pulmonary function improved during this period, pulmonary function was still significantly lower in patients with PCLI compared to patients without PCLI.

Pulmonary function did not improve significantly between four and fourteen hours. This result was consistent with previous studies in patients who underwent CABG [17] and aortic arch replacement [14]. Most patients, even those with PCLI, fulfilled the criteria for extubation, including a PFR ≥ 200 or even less [21]. Thus, continuing intubation was not due to pulmonary function. Our actual intubation rate was 45.4% at 8 h and 38.3% at 12 h, even in patients without postoperative adverse events. Nevertheless, PFR and RI at 2 h from CPB cessation were strong predictors of postoperative DMV. These findings indicate that PFR and RI two hours after CPB may be linked to cardiac performance, renal function, hemodynamic status, and other factors influencing extubation criteria. Further investigations are necessary to elucidate the relationships between post-CPB pulmonary function and extubation criteria.

Limitations of the Study

The present study had several limitations. Firstly, this was a retrospective study; therefore, blood samples were not always collected on time and the variation of measurement times might have affected the results. Secondly, perioperative management, including ventilator settings, was not the same in all patients, although we worked as one team under the same policy and strategy during the study period. Finally, all patients were ethnically Japanese and were treated by a single surgical team, which may have introduced ethnic and team-based biases.

## 5. Conclusions

PFR and RI at two hours after CPB were correlated with postoperative DMV and lengths of ICU and hospital stays. PFR ≥ 354 and RI ≤ 0.83 satisfactorily predicted early extubation, and PFR < 213 and RI > 1.68 predicted prolonged DMV. The management of patients with PCLI should be the next target to improve early outcomes in contemporary valve surgery.

## Figures and Tables

**Figure 1 jcm-10-03262-f001:**
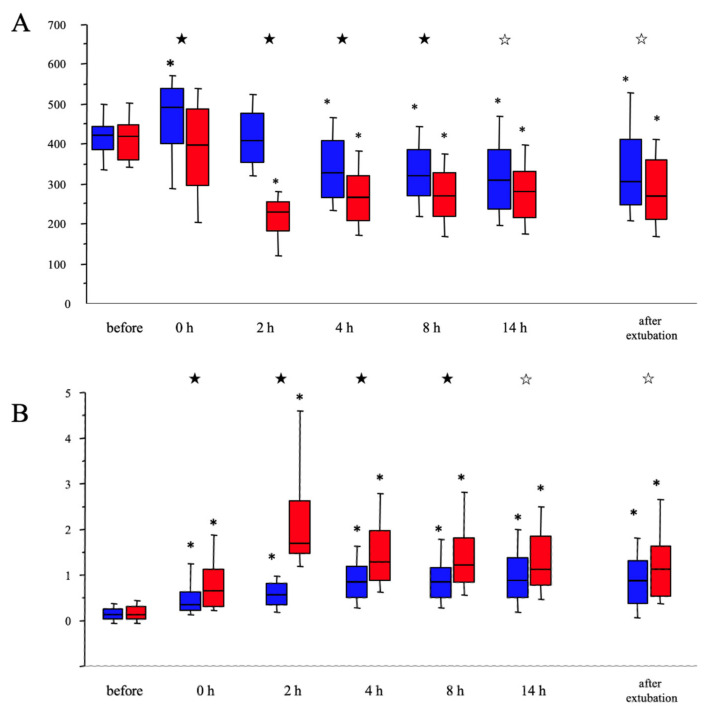
Time-dependent changes in PaO_2_/FiO_2_ (PFR) and respiratory index (RI) in patients with and without post-cardiopulmonary bypass (CPB) lung injury (PCLI). (**A**) Time-dependent changes in PFR. PFR was significantly decreased after CPB in patients with PCLI. PFR was higher in patients without PCLI as compared to patients with PCLI throughout the observation period, except before surgery. PFRs at 4, 8, and 12 h after surgery and after extubation were significantly decreased compared with PFRs in the preoperative period. (**B**) Time-dependent changes in RI. RI was significantly increased after CPB in both groups. RI was always higher in patients with PCLI compared to RI in patients without PCLI. *^1^, *p* < 0.05 vs. before surgery; *^2^, *p* < 0.05 vs. before surgery; ★ *p* < 0.001 between patients with and without PCLI; ☆, *p* < 0.01 between patients with and without PCLI.

**Figure 2 jcm-10-03262-f002:**
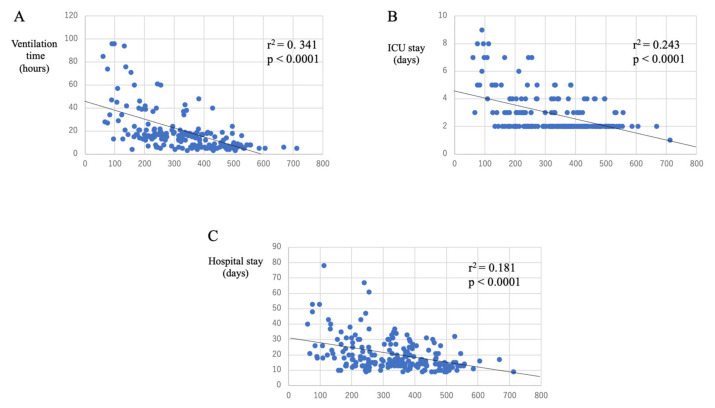
Correlations between PaO_2_/FiO_2_ (PFR) and postoperative course. PFR is positively correlated with DMV (**A**), ICU stays (**B**), and hospital stays (**C**). DMV, duration of mechanical ventilation; ICU, intensive care unit.

**Figure 3 jcm-10-03262-f003:**
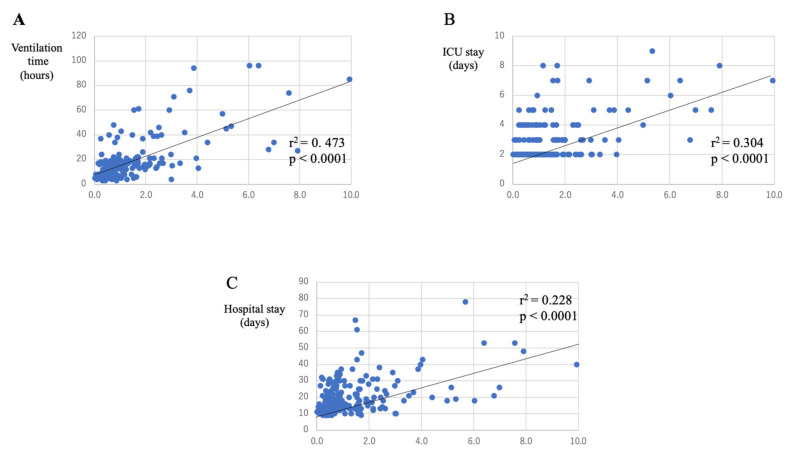
Correlations between respiratory index (RI) and postoperative course. RI is positively correlated with DMV (**A**), ICU stays (**B**), and hospital stays (**C**). DMV, duration of mechanical ventilation; ICU, intensive care unit.

**Figure 4 jcm-10-03262-f004:**
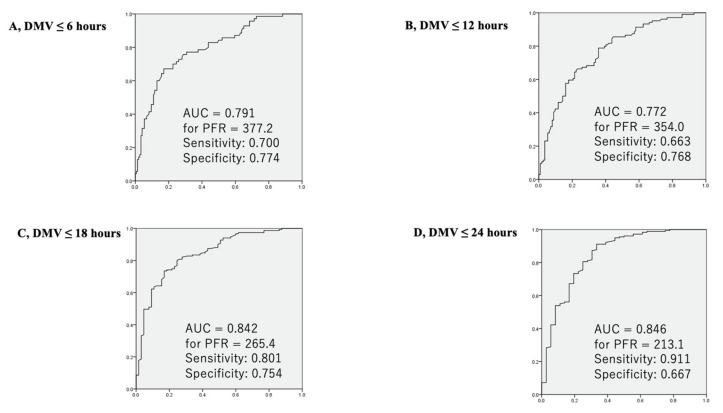
Receiver operating characteristic (ROC) curves for PaO_2_/FiO_2_ (PFR) as a predictor of the duration of mechanical ventilation shorter than 6 h (**A**), 12 h (**B**), 18 h (**C**), and 24 h (**D**) are presented. (**A**) PFR > 377.2 showed the highest sensitivity (0.700) and specificity (0.774) with an area under the curve of 0.791. (**B**) PFR > 354.0 showed the highest sensitivity (0.663) and specificity (0.768) with an area under the curve of 0.772. (**C**) PFR > 265.4 showed the highest sensitivity (0.801) and specificity (0.754) with an area under the curve of 0.842. (**D**) PFR > 213.1 showed the highest sensitivity (0.911) and specificity (0.667) with an area under the curve of 0.846.

**Figure 5 jcm-10-03262-f005:**
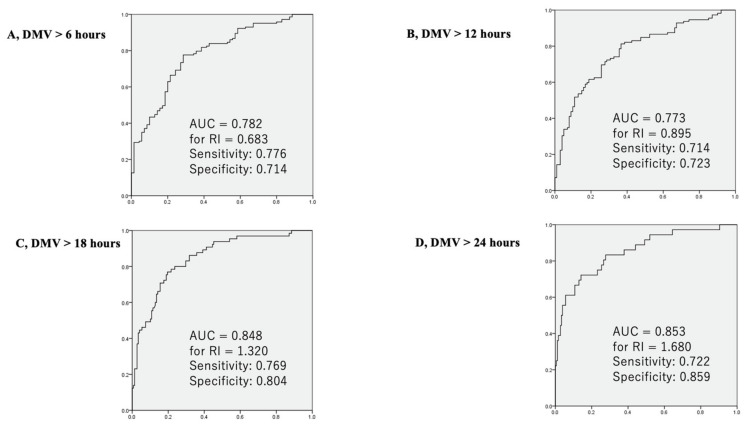
Receiver operating characteristic (ROC) curves for the respiratory index as a predictor of the duration of mechanical ventilation shorter than 6 h (**A**), 12 h (**B**), 18 h (**C**), and 24 h (**D**) are presented. (**A**) RI < 0.683 showed the highest sensitivity (0.776) and specificity (0.714) with an area under the curve of 0.782. (**B**) RI < 0.895 showed the highest sensitivity (0.714) and specificity (0.723) with an area under the curve of 0.773. (**C**) RI < 1.320 showed the highest sensitivity (0.769) and specificity (0.804) with an area under the curve of 0.848. (**D**) RI < 1.680 showed the highest sensitivity (0.722) and specificity (0.859) with an area under the curve of 0.853.

**Table 1 jcm-10-03262-t001:** Baseline characteristics.

	Whole Cohort	PFR *^7^ ≥ 300	PFR < 300	*p*-Value *^8^
Number of patients	225	137	88	
Age (years)	67.7 ± 13.16	66.6 ± 14.22	69.3 ± 11.20	0.299
Octogenarian (%)				
Male (%)	154 (68.4%)	91 (66.4%)	63 (71.6%)	0.464
Body mass index	22.9 ± 3.99	22.3 ± 3.85	23.8 ± 4.07	0.017
Previous smoking (%)	119 (52.9%)	67 (48.9%)	52 (59.1%)	0.171
Current smoking (%)	33 (14.7%)	17 (12.4%)	16 (18.2%)	0.251
Diabetes mellitus (%)	52 (23.1%)	25 (18.2%)	27 (30.7%)	0.036
Hypertension (%)	170 (75.6%)	97 (70.8%)	73 (83.0%)	0.040
Hyperlipidemia (%)	135 (60.0%)	78 (56.9%)	57 (64.8%)	0.266
Hemodialysis (%)	16 (7.1%)	4 (2.9%)	12 (13.6%)	0.003
eGFR *^1^	60.13 ± 26.20	65.07 ± 26.01	56.18 ± 27.53	0.039
Previous stroke (%)	29 (12.9%)	13 (9.5%)	16 (18.2%)	0.068
Hemoglobin (g/dL)	12.7 ± 3.94	12.6 ± 1.96	12.9 ± 5.82	0.245
Platelet (/mL)	203.5 ± 83.64	208.8 ± 91.16	195.3 ± 69.99	0.472
NYHA ≥ II (%)	160 (71.1%)	95 (69.3%)	65 (73.8%)	0.547
Peripheral arterial disease	12 (5.3%)	4 (2.9%)	8 (9.1%)	0.066
LVEF (%)	62.5 ± 18.6	62.8 ± 14.20	62.0 ± 23.99	0.116
Pulmonary hypertension *^2^ (%)	47 (20.9%)	25 (18.2%)	22 (25.0%)	0.242
COPD *^3^ (%)	67 (29.8%)	38 (27.7%)	29 (33.0%)	0.455
PO_2_ *^4^ (mmHg)	85.9 ± 14.51	88.2 ± 14.11	82.2 ± 15.18	0.337
PCO_2_ *^4^ (mmHg)	39.0 ± 5.23	39.8 ± 5.18	37.8 ± 5.34	0.782
%VC *^5^	86.4 ± 18.56	90.9 ± 18.47	79.4 ± 18.78	0.659
FEV1% *^6^	88.2 ± 23.64	92.5 ± 23.77	81.4 ± 23.56	0.655

*^1^: Estimated glomerular filtration rate. *^2^: diagnosed in preoperative echo cardiogram or catheter examinations. *^3^: chronic obstructive pulmonary disease. *^4^: values are under room air atmosphere. *^5^: vital capacity/estimated vital capacity. *^6^: forced expiratory volume in one second/vital capacity. *^7^: PO_2_/FiO_2_ at 2 h after cardiopulmonary bypass. *^8^: *p*-value between PFR ≥ 300 and <300.

**Table 2 jcm-10-03262-t002:** Intraoperative parameters and postoperative course.

	Whole Cohort	PFR ≥ 300	PFR *^9^ < 300	*p*-Value *^10^
***(Type of operation)***				
Mitral valve surgery	96 (42.7%)	68 (49.6%)	28 (31.8%)	0.009
Aortic valve surgery	106 (47.1%)	64 (46.7%)	42 (47.7%)	0.892
Mitral + Aortic valves	23 (10.2%)	5 (3.6%)	18 (20.5%)	<0.001
Additional procedures	118 (52.4%)	65 (47.4%)	53 (60.2%)	0.075
*Tricuspid repair*	47 (20.9%)	34 (24.8%)	13 (14.8%)	0.092
*CABG* *^1^	52 (23.1%)	26 (19.0%)	26 (29.5%)	0.076
*Maze*	34 (15.1%)	15 (10.9%)	19 (21.6%)	0.036
*Emergency*	15 (6.7%)	5 (3.6%)	10 (11.4%)	0.003
*Re-sternotomy*	5 (2.2%)	1 (1.1%)	4 (2.9%)	0.651
***(Intraoperative parameters)***				
Operation time (min)	305.3 ± 107.18	281.3 ± 89.4	342.7 ± 119.1	<0.001
CPB *^2^ time (min)	166.1 ± 68.52	147.2 ± 50.4	195.4 ± 80.4	<0.001
ACC *^3^ time (min)	101.7 ± 38.0	96.2 ± 33.4	110.4 ± 41.9	<0.001
Fentanyl	17.4 ± 4.5	17.2 ± 4.8	17.7 ± 4.3	0.463
RBC *^4^ transfusion (unit)	99 (44.0%)	50 (36.5%)	49 (55.7%)	0.003
FFP *^5^ transfusion (unit)	74 (32.9%)	38 (27.7%)	36 (40.9%)	0.040
Platelet transfusion (%)	33 (14.7%)	10 (7.3%)	23 (26.1%)	<0.001
Water balance (mL)	2560 ± 2080	2148 ± 1655	3200 ± 2447	<0.001
lowest hemoglobin *^6^	6.8 ± 1.23	6.9 ± 1.27	6.6 ± 1.17	0.516
***(Postoperative parameters)***				
Inotrope ≥ medium *^7^	44 (19.6%)	24 (17.5%)	20 (22.7%)	0.305
Transfusion *^8^	21 (9.3%)	9 (6.6%)	12 (13.6%)	0.097
Adverse events (%)	41 (18.2%)	16 (11.7%)	25 (28.4%)	0.001
Ventilation time (h)	18.9 ± 22.75	11.4 ± 7.9	30.5 ± 31.78	<0.001
ICU stay (days)	3.2 ± 5.47	2.9 ± 6.62	3.6 ± 2.72	<0.001
Hospital stay (days)	20.0 ± 17.47	18.4 ± 19.23	22.6 ± 13.72	<0.001
Hospital death (%)	0.9	0.0	2.3	0.152

*^1^: coronary artery bypass graft; *^2^: cardiopulmonary bypass; *^3^: aortic cross-clamp; *^4^: red blood cell; *^5^: frozen fraction of plasma; *^6^: the lowest value during cardiopulmonary bypass; *^7^: dopamine + dobutamine ≥ 5 μg/kg/min or norepinephrine ≥ 0.15 μg/kg/min; *^8^: transfusion of red blood cells > 4 units or frozen fraction of plasma > 4 units or platelets during surgery; *^9^: PO_2_/FiO_2_ at 2 h after cardiopulmonary bypass; *^10^: *p*-value between PFR ≥ 300 and <300.

**Table 3 jcm-10-03262-t003:** Correlations between postoperative pulmonary function and clinical outcomes.

	Ventilation Time		ICU *^4^ Stay		Hospital Stay	
Time after CPB *^1^	Coefficient (r^2^)	*p*-Value	Coefficient (r^2^)	*p*-Value	Coefficient (r^2^)	*p*-Value
(immediate)						
PFR *^2^	0.104	<0.001	0.038	0.006	0.034	0.011
RI *^3^	0.112	<0.001	0.067	0.001	0.071	<0.001
(2 h)						
PFR	0.294	<0.001	0.145	<0.001	0.126	<0.001
RI	0.369	<0.001	0.213	<0.001	0.121	<0.001
(4 h)						
PFR	0.085	<0.001	0.045	0.002	0.053	0.001
RI	0.181	<0.001	0.091	0.006	0.061	<0.001
(8 h)						
PFR	0.102	<0.001	0.074	<0.001	0.060	<0.001
RI	0.264	<0.001	0.153	<0.001	0.061	<0.001

*^1^: cardiopulmonary bypass, *^2^: PO_2_/FiO_2_, *^3^: respiratory index, *^4^: intensive care unit.

## Data Availability

Data can be made available on request from established research groups with an appropriate data-sharing agreement. Please contact the corresponding author for data sharing.

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
