# Peer review of "Postoperative Changes in Pulmonary Function after Valve Surgery: Oxygenation Index Early after Cardiopulmonary Is a Predictor of Postoperative Course"

_jcm, 2021, doi:10.3390/jcm10153262_

Round 1

Reviewer 1 Report

Thank you for the opportunity to review your manuscript. I have reviewed the manuscript entitled " Postoperative changes in pulmonary function after valve surgery: oxygenation index early after cardiopulmonary is a predictor of postoperative course". This report provides excellent data of the lung injury after complexed valve surgery. In many previous reports, the amount of bleeding, operation time, CPB time et.al. have been revealed as the risk factors of a bad clinical course after cardiac surgery. The authors found post-CPB lung injury is a direct predictor of the adverse postoperative course. This is a very attractive report for clinicians.    [Major comments] (1) Table 1: the cut off value of PFR The cut off value of PFR was more than 300 or not in table 1. Why did the authors set this value? The cut off value of PFR of early extubation, less than 12 hours,  was about 350 in this manuscript. I recommend the authors to change the cut off value in table 1.   [Minor comments] (1) statistical analysis The authors analyzed the risk factors of DMV with univariate and multivariate analysis. But the methods of them were not written in the method section. Please describe the analytic methods.   (2) typographical error Table: Body mass indexI: 'Body mass index' is correct.

Author Response

(Reply to Reviewer #1)

Thank you very much for reviewing our manuscript. We greatly appreciate for your valuable comments. We thoroughly checked and revised our manuscript in accordance with them. We hope the new manuscript will satisfy your concerns.

[Major comments] (1) Table 1: the cut off value of PFR. The cut off value of PFR was more than 300 or not in table 1. Why did the authors set this value? The cut off value of PFR of early extubation, less than 12 hours, was about 350 in this manuscript. I recommend the authors to change the cut off value in table 1.  

Thank you very much for this comment. The reason we used PFR300 was basically from the American-European Consensus Conference definition (Am J Respir Crit Care Med.1994;149(3 pt 1):818-824) and the Berlin definition of ARDS (JAMA.2012;307:2526-2533, 2018;319:698-710). In both, the border line between normal and lung injury was at PFR300. This setting has been followed in many studies for post-cardiopulmonary lung injury (PCLI) so far, and this is why we adopted this definition in the present study. As we said in the introduction, one of our purposes of this study was to investigate “whether PCLI significantly influences the postoperative course after current valve surgery”. And therefore, we divided patients by using PFR300, and successfully showed patients with PCLI had worse clinical outcomes. This definition is also used in Figure-1 in which the different changing pattern of pulmonary function is successfully shown. Because the impact of PFR as a predictor of ventilation time was greater than this finding, the spot light might not be on it enough in the manuscript, but if we used PFR354 for the tables, some may say it should be 377 (indicating ventilation time <6h) and others can say “why it is not 213 (this is for ventilation time >24h)?”. We are off course happy to use PFR354 but hopefully we would still like to stick to PFR300 due to above reason. We attached the tables using PFR354 below to show you how it looks like. Again, if you think we should use the table below, we can do it.

Table 1 Baseline characteristics and type of surgery

Whole cohort

PFR*7≥354

PFR<354

p*8 value

Number of patients

225

104

121

Age

67.7±13.16

65.3±14.52

69.7±11.54

0.040

Octogenarian (%)

Male (%)

154 (68.4%)

66 (63.5%)

88.0 (72.7%)

0.136

BMI

22.9±3.99

22.4±4.12

23.3±3.84

0.057

Previous smoking (%)

119 (52.9%)

51.0 (49%)

68.0 (56.2%)

0.283

Current smoking (%)

33 (14.7%)

16.0 (15.4%)

17.0 (14.0%)

0.778

Diabetes Melitus (%)

52 (23.1%)

21.0 (20.2%)

31.0 (25.6%)

0.336

Hypertension (%)

170 (75.6%)

69 (66.3%)

101 (83.5%)

0.003

Hyperlipidemia (%)

135 (60.0%)

59 (56.7%)

76 (62.8%)

0.353

Hemodialysis (%)

16 (7.1%)

1 (1.0%)

15 (12.4%)

0.001

Serum creatinine*1

1.5±2.16

0.9±0.63

1.9±2.81

<0.001

eGFR*1

60.1±26.2

68.1±23.5

53.3±26.5

<0.001

Previous stroke (%)

29 (12.9%)

10.0 (9.6%)

19.0 (15.7%)

0.174

Hb

12.7±3.94

12.7±1.88

12.8±5.09

0.118

PLT

203.5±83.64

215.2±96.85

193.5±69.03

0.065

NYHA ≥II (%)

160 (71.1%)

75 (72.1%)

85 (70.2%)

0.758

Peripheral arterial disease

12 (5.3%)

3 (2.9%)

9 (7.4%)

0.130

LVEF

62.5±18.6

62.8±15.22

62.1±21.14

0.073

Pulmonary hypertension*2 (%)

47 (20.9%)

17 (16.3%)

30 (24.8%)

0.120

COPD

67 (29.8%)

25 (24%)

42 (34.7%)

0.090

PO2*3 (mmHg)

85.9±14.51

88.9±15.47

83.3±13.52

0.176

PCO2*4 (mmHg)

39.0±5.23

39.3±4.97

38.8±5.44

0.416

%VC*5

86.4±18.56

90.2±18.44

83.1±18.67

0.518

FEV1%*6

88.2±23.64

91.8±24.88

85.0±22.53

0.710

Table 2.  Intraoperative parameters and postoperative course

Whole cohort

PFR≥354 (n=104)

PFR<354 (n=121)

p value

(Type of operation)

Mitral valve surgery

96 (42.7%)

53 (51%)

43 (35.5%)

0.020

Aortic valve surgery

106 (47.1%)

48 (46.2%)

58 (47.9%)

0.790

Mitral + Aortic

23 (10.2%)

3 (2.9%)

20 (16.5%)

0.001

Additional procedures

118 (52.4%)

45 (43.3%)

73 (60.3%)

0.011

Tricuspid repair

47 (20.9%)

24 (23.1%)

23 (19.0%)

0.454

CABG*3

52 (23.1%)

18 (17.3%)

34 (28.1%)

0.056

Maze

34 (15.1%)

11 (17.3%)

23 (19.0%)

0.078

Emergency

15 (6.7%)

5 (4.8%)

10 (8.3%)

0.300

Re-sternotomy

5 (2.2%)

1 (1.0%)

4 (3.3%)

0.234

(intraoperative parameters)

Operation time (min)

305.3±107.18

272.7±84.2

333.4±115.5

<0.001

CPB*1 time (min)

166.1±68.52

144.0±45.9

185.0±77.9

<0.001

ACC*2 time (min)

101.7±38.0

93.4±32.9

108.9±39.7

<0.001

Use of total CPB (%)

RBC*3 transfusion (%)

99 (44.0%)

32 (30.8%)

67 (55.4%)

<0.001

FFP*4 transfusion (%)

74 (32.9%)

22 (21.2%)

52 (43%)

<0.001

Platelet transfusion (%)

33 (14.7%)

5 (4.8%)

28 (23.1%)

<0.001

Water balance (ml)

2560±2080

2063±1686

2987±2265

<0.001

      CPB balance

1290±1671

1101±1455

1452±1828

0.200

lowest hemoglobin*5

6.8±1.23

6.9±1.25

6.7±1.21

0.492

(postoperative parameters)

Inotrope ≥ medium*6

44(19.6%)

14 (13.5%)

30 (24.8%)

0.026

Transfusion*7

21(9.3%)

5 (4.8%)

16 (13.2%)

0.026

Adverse events

41(18.2%)

9 (8.7%)

32 (26.4%)

<0.001

Ventilation time (hours)

18.9±22.75

9.9±7.0

26.6±28.1

<0.001

ICU stay (days)

3.2±5.47

3.0±7.59

3.3±2.43

<0.001

Hospital stay (days)

20.0±17.47

17.8±21.56

22.0±12.44

<0.001

Hospital death (%)

2 (0.9%)

0 (0.0%)

2 (1.65%)

0.501

[Minor comments] (1) statistical analysis. The authors analyzed the risk factors of DMV with univariate and multivariate analysis. But the methods of them were not written in the method section. Please describe the analytic methods.  

We are very sorry. We added the description as below in the section of statistical analysis.

Univariate and multivariate logistic regression analyses were performed to identify predictors of prolonged DMV. The Hosmer-Lemeshow test was performed to evaluate the validity of the multivariate logistic regression model.

(2) typographical error Table: Body mass indexI: 'Body mass index' is correct.

Thank you very much for indicating our mistake. We revised.

Reviewer 2 Report

Comment 1

L46-66

In the introduction, the authors write that “complex valve surgery requires longer CPB”. In the methods section, the authors state that the study population underwent aortic or mitral valve surgery. What is the definition of “complex” in this context? Simple isolated aortic valve replacement does not require long CPB. Please explain.

Comment 2

L75

Why do the authors routinely perform bicaval cannulation? Please clarify.

Comment 3

L108

The authors defined PCLI as Horovitz Index <300. However, this is often considered as only mild injury. Moderate injury is defined as a ratio of 200 to 300, and severe injury as a ratio <100. How many patients had mild, moderate and severe lung injury? Were there any differences in the postoperative course between the three groups?

Comment 4

L127 and table 1

The was no statistically significant difference with regards to smoking between the two groups (p>0.05). In turn, the authors statement in the results section (L127) is not correct. Please revise.

Comment 5

Tables 1 and 2

In table one, the authors show the pulmonary lung injury group in column 4, while the same group is shown in column 3 in table 2. This is confusing and makes it harder for the reader to understand. Please unify.

Comment 6

L143 – 147 and table 3

Table 3 contains a lot of data and is quite confusing. I suggest to delete the entire univariate analysis from the main article, and present it as supplementary data instead. Further, I suggest to present only the statistically significant variables of the multivariate analysis in the main article (PCLI and transfusion). All other non-significant variables should be shown in the supplementary data.

Comment 7

L288

The authors acknowledge that blood samples were not always collected on time. I suggest adding the mean times of collection plus standard deviation.

Comment 8

All tables, entire article

The authors present percentages only. Please provide numbers (n) and percentages. This is more appropriate.

Author Response

(Reply to Reviewer #2)

Thank you very much for reviewing our manuscript. We carefully read your comments and revised the manuscript as below. We believe the manuscript is now much better than before. We hope our new manuscript will meet your satisfaction.

Comments and Suggestions for Authors

Comment 1

L46-66

In the introduction, the authors write that “complex valve surgery requires longer CPB”. In the methods section, the authors state that the study population underwent aortic or mitral valve surgery. What is the definition of “complex” in this context? Simple isolated aortic valve replacement does not require long CPB. Please explain.

Thank you very much for this comment. The word “complex” was used to express our whole cohort because more than a half of our patients had 2 or more additional procedures over aortic or mitral surgery. As a result, our CPB time and Euroscore II were much longer and higher than those in previous studies (mean CPB time was about 167 min and Euroscore II was 4.2, although median values presented in the manuscript were lower than them). But as you indicated, the word “complex” might make a confusion because this study also included single valve surgeries like AVR, off course. We deleted the word “complex” in the 3rd paragraph of introduction, the first sentence of discussion and conclusion. We believe it is now clear for readers. Thank you very much again for telling us about this confusion.

Comment 2

L75

Why do the authors routinely perform bicaval cannulation? Please clarify.

Thank you very much for this question. Long time ago, we used to do AVR using single venous drainage via right atrial appendage. However, we experienced severe bleeding from the aortic root on weaning from CPB after AVR, and bicaval drainage was more suitable for the repair (to decompress the RV outflow and have an optimal surgical view at the aortic root) at that time. Since then, we have been using bicaval veinous tubes for all AVR and now it is routine for us. For mitral surgeries we use bicaval drainage as most of surgeons do. As you know it is useful to control the blood flow from pulmonary veins. As a result, we now do all valve surgeries using bicaval cannulation.

Comment 3

L108

The authors defined PCLI as Horovitz Index <300. However, this is often considered as only mild injury. Moderate injury is defined as a ratio of 200 to 300, and severe injury as a ratio <100. How many patients had mild, moderate and severe lung injury? Were there any differences in the postoperative course between the three groups?

Thank you very much for this comment. The table below shows the differences in postoperative course among patients with different degrees of lung injury. Although the data are shown in mean±SD, statical analysis was done using non-parametric test including Kruskal-Wallis test for the first analysis and each inter-group analysis followed with Bonferoni correction. As table shows, the grade of lung injury is well correlated to postoperative course. Since we already showed this correlation in Figure-2 and 3, we did not think this table was necessary, but we are happy to add this table or some comments if you think necessary. Thank you very much again, for this comment.

PFR≥300

300>PFR≥200

200>PFR≥100

100>PFR

Patient numbers

137

58

22

8

Ventilation*1

11.4±7.9*2

12.8±17.7

34.5±23.8

80.1±69.8*4

ICU stay*1

2.9±6.6*2

3.5±3.0

3.6±1.7

5.4±2.3*4

Hosp Stay*1

18.4±19.2*3

22.9±16.2

24.4±9.5

25.3±14.3

*1: p<0.05 among 4 groups (PFR≥300, 300>PFR≥200, 200>PFR≥100, 100>PFR), *2: p<0.05 vs other 3 groups, *3: p<0.05 vs 200>PFR≥100 and 100>PFR, *4: p<0.05 vs 300>PFR≥200

Comment 4

L127 and table 1

The was no statistically significant difference with regards to smoking between the two groups (p>0.05). In turn, the authors statement in the results section (L127) is not correct. Please revise.

We are very sorry for this mistake. We revised.

Comment 5

Tables 1 and 2

In table one, the authors show the pulmonary lung injury group in column 4, while the same group is shown in column 3 in table 2. This is confusing and makes it harder for the reader to understand. Please unify.

Thank you very much for this comment. We revised Table 1 and 2

Comment 6

L143 – 147 and table 3

Table 3 contains a lot of data and is quite confusing. I suggest to delete the entire univariate analysis from the main article, and present it as supplementary data instead. Further, I suggest to present only the statistically significant variables of the multivariate analysis in the main article (PCLI and transfusion). All other non-significant variables should be shown in the supplementary data.

We excluded Table 3 from the manuscript and it is now presented as “Supplementary Table 1” in the revised manuscript. The data of multivariate analysis on PCLI and transfusion are described in the result section. Thank you very much for this comment.

Comment 7

L288

The authors acknowledge that blood samples were not always collected on time. I suggest adding the mean times of collection plus standard deviation.

Thank you very much for this comment. Unfortunately, we did not record the actual sample measurement time. When we collected blood gas analysis data from medical records, we just checked the time and wrote down if it was within 20 min or not (simply as “yes” or “no”). Therefore, we cannot show the mean times of collection. We are very sorry about this, but still believe our results had strong impact for readers. We added below sentence in “limitation of the study”.   

The variation of measurement time might affect the results.

Comment 8

All tables, entire article

The authors present percentages only. Please provide numbers (n) and percentages. This is more appropriate.

Thank you very much for indicating this point. We added numbers in Tables and result section in the text.

Round 2

Reviewer 2 Report

Thank you for the revision. The article's quality has improved.